# Satellite Sea Surface Temperature Product Comparison for the Southern African Marine Region

Matthew Carr [1,*], Tarron Lamont [2,3,4] and Marjolaine Krug [2,4] 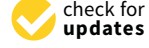

1   South African Environmental Observation Network (SAEON), Cape Town 8012, South Africa
2   Department of Environment, Forestry and Fisheries, Oceans and Coasts Research,
    Cape Town 8012, South Africa; tlamont@environment.gov.za (T.L.); Mkrug@environment.gov.za (M.K.)
3   Bayworld Centre for Research & Education, Cape Town, South Africa South African Environmental
    Observation Network (SAEON), Cape Town 7806, South Africa
4   Department of Oceanography, Nansen-Tutu Centre for Marine Environmental Research,
    University of Cape Town, Cape Town 7701, South Africa
*   Correspondence: matt@saeon.ac.za

**Abstract:** Several satellite-derived Sea Surface Temperature (*SST*) products were compared to determine their potential for research and monitoring applications around the southern African marine region. This study provides the first detailed comparison for the region, demonstrating good overall agreement (variance < 0.4 °C$^2$) between merged *SST* products for most of the South African marine region. However, strong disagreement in absolute *SST* values (variance of 0.4–1.2 °C$^2$ and differences of up to 6 °C) was observed at well-known oceanographic features characterized by complex temperature structures and strong *SST* gradients. Strong seasonal bias in the discrepancy between *SST* was observed and shown to follow seasonal increases in cloud cover or local oceanographic dynamics. Disagreement across the L4 products showed little dependence on their spatial resolutions. The periods of disagreement were characterized by large deviations among all products, which resulted mainly from the lack of input observations and reliance on interpolation schemes. This study demonstrates that additional methods such as the ingestion of additional in situ observations or daytime satellite acquisitions, especially along the west coast of southern Africa, might be required in regions of strong *SST* gradient, to improve their representations in merged *SST* products. The use of ensemble means may be more appropriate when conducting research and monitoring in these regions of high *SST* variance.

**Keywords:** sea surface temperature; southern Africa; Benguela upwelling system; Agulhas Current system; satellite oceanography; remote sensing

## 1. Introduction

Sea surface temperature (*SST*) data are essential to environmental research, monitoring, and forecasting. *SST* fields are extensively used in oceanographic and atmospheric research to identify and investigate oceanographic processes, air-sea interactions, and long-term climate variability [1–5]. *SST* fields are also vital inputs for the numerical ocean models used by weather and operational oceanography forecasting systems, which in turn inform industry, government agencies and the general public [6]. Ocean temperature also strongly influences the distribution and diversity of biota as well as the functioning of ecosystems, rendering *SST* an essential variable when monitoring the impacts of climate change in environmentally sensitive systems [7,8].

Some of the key oceanographic processes within the southern African region that are identified and monitored using *SST* data include the Agulhas Current frontal variability (at both the meso and sub-meso scales), Agulhas Rings or Agulhas Current intrusions, marine heat waves, marine cold-spells, coastal upwelling and large-scale climate modes, such as Benguela Niños [9–12]. These processes strongly impact the marine ecosystem

and the local climate system. Marine heat waves have been shown to negatively impact commercial variable fisheries, marine mammals and seabirds [13,14]. Similarly, Benguela Niños have been shown to result in severe reductions in pelagic and benthic organisms as well as significant increases in local rainfall events [15,16]. Intensification in the coastal upwelling systems may result in shifts in pelagic populations [17].

The majority of *SST* data sets used in oceanographic research and operational systems are derived from satellite observations using either thermal infrared (IR) or passive microwave (MW) sensors. The IR and MW instruments have unique sampling characteristics. MW radiometers are able to penetrate cloud cover with little attenuation providing spatially complete *SST* fields. However, MW retrievals are limited to a relatively low spatial resolution of between 25 and 50 km due to the ratio of the radiation wavelength to the antenna diameter [18–20]. Furthermore, the accuracy of MW retrievals is reduced within ~50 km from the coast due to land contamination and radio frequency interferences, resulting in a coastal band of missing data in MW-derived *SST* data sets [21]. Conversely, IR retrievals are strongly impacted by aerosols, resulting in large amounts of missing data due to atmospheric particles such as cloud cover, but are capable of achieving high spatial resolution ±1 km [22]. In order to compensate for the shortcomings of the respective sampling capabilities, IR and MW retrievals are merged using various interpolation techniques to provide spatially complete, high resolution *SST* fields [23,24]. The merged and interpolated data sets, referred to as Level 4 (L4) *SST* products, have become increasingly important for environmental research and operational systems, which in turn has led to the development of numerous freely accessible L4 *SST* products [6,24,25]. The near real time availability of L4 *SST* products allows for the near real time monitoring of the ocean's surface temperature and dynamics, with a more complete spatial coverage than is possible from individual L3 products, and may at times enable the prediction of oceanic variability. Ref. [26] used a L4 *SST* product to monitor the location of the Agulhas Current front during the Shelf Agulhas Glider Experiment. L4 *SST* products could also be used to predict climate mode variability such as Benguela Niños which may allow for adaptive strategies against the negative effects on marine ecosystems [27].

The various L4 *SST* products each use different combinations of source data, which may also include in situ data, as well as various merging and interpolation techniques. The inherent differences between source data and processing methods can, in turn, result in large regional differences in the accuracy of L4 *SST* products [24,25,28]. Considering the large range of available *SST* products and their intrinsic differences, it is imperative that the performance of *SST* products is compared, since poorly performing products may strongly impact research or operational outputs [20].

While the performance of L4 *SST* products has been compared globally and for some specific regions [24,25,28–30], there have been relatively few studies focused on the suitability of various *SST* products around southern Africa, with little to no focus on comparing the performance of L4 *SST* products. Studies that have been conducted around southern Africa have shown discrepancies between single sensor *SST* products (Level 3) in the nearshore regions. A warm bias of 3–5 °C in the Pathfinder *SST* compared to MODIS Terra *SST* was observed within the Benguela upwelling system [31,32], while biases of up to 6 °C between in situ coastal temperatures and these satellite products were reported in nearshore regions around South Africa [33].

A regionally focused assessment of the L4 *SST* products around the southern African marine environment is thus needed as the region is characterized by numerous oceanographic processes and features, each with distinctive, complex *SST* structures. The west coast of southern Africa is dominated by the strong, wind-driven, coastal Benguela upwelling system (Figure 1) which results in substantial offshore *SST* gradients of about 3 °C/100 km [34]. This upwelling system is typically divided into distinct northern and southern regions based on oceanographic and biological dynamics [35,36]. The northern Benguela system (15–29° S) is characterized by perennial upwelling, while the southern Benguela (29–34.5° S) has a strong seasonality with increased upwelling favorable winds

in the austral summer months [35]. Correctly capturing changes in *SST* in the Benguela coastal upwelling regions and the associated *SST* fronts is important as variability in the upwelling dynamics is likely to strongly impact on the local marine ecosystem [12,37–39]. The Benguela upwelling system is bound by two warm ocean currents, the Angola Current in the north and Agulhas Current to the south [35,40,41]. The juxtaposition of the Benguela upwelling and Angola Current forms the near permanent Angola–Benguela front between 14–16° S [35]. The Angola–Benguela front is a relatively stable *SST* feature, characterized by seasonal positional shifts, with the front being furthest south during the austral summer. In contrast, the southern boundary of the Benguela upwelling system and Agulhas Current is extremely dynamic, characterized by eddies and filaments which leak from the retroflection of the Agulhas Current [42].

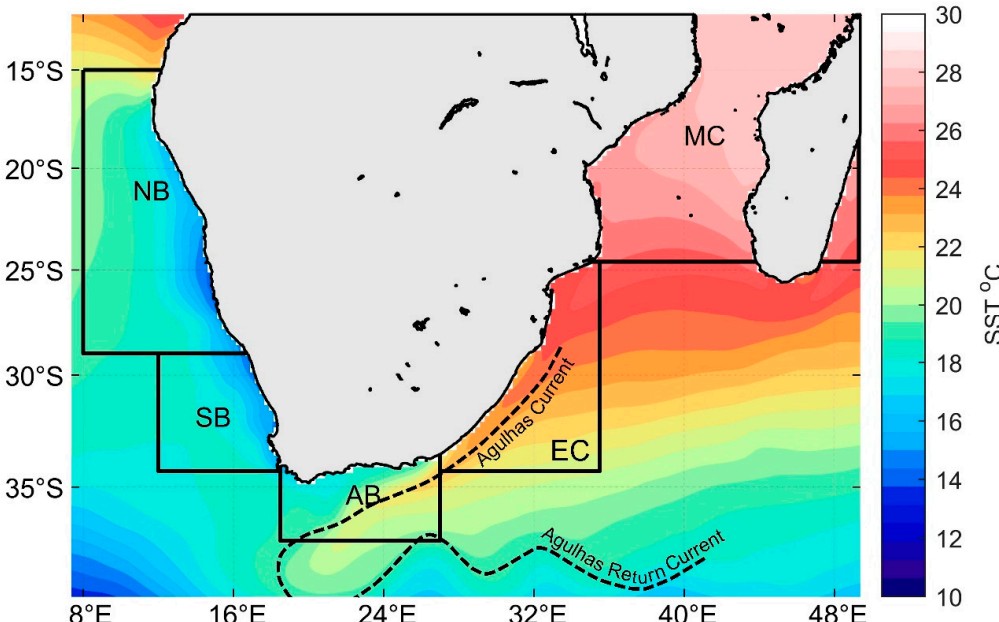

**Figure 1.** Mean Sea surface temperature *(SST)* from AVHRR_OI*SST* (2 September 1981 to 23 April 2019). The black boxes represent the five sub regions. The abbreviations NB, SB, AB, EC and MC indicate the northern Benguela, southern Benguela, Agulhas Bank, East Coast and Mozambique Channel, respectively. The dashed black line is a schematic of the Agulhas Current system depicting the Agulhas Current and Agulhas Return Current.

*SST* characteristics along the east coast of South Africa (24.6–37.3° S) are dominated by the Agulhas Current system and its distinct thermal signature (Figure 1). Strong *SST* gradients of 5 to 10 °C/100 km are observed at the Agulhas Current front [43]. The strong thermal gradients at the inshore edge of the Agulhas Current have been used in the past to track the occurrence and propagation of large Agulhas Current meanders and provide new insight on the Agulhas Current mesoscale variability [44]. Accurate *SST* observations at the inshore edge of Agulhas Current are also needed to understand and predict Air/Sea interaction processes in this region, as the magnitude of the frontal gradient drives a distinct atmospheric response with sudden increases in wind speeds and heat fluxes over the Agulhas Current's front and its warm core [43,45]. *SST* variability inshore of the Agulhas Current is driven in part by the Agulhas Current's frontal instability and in part by wind-driven coastal upwelling. Frontal variability at the inshore edge of the Agulhas Current occurs over a range of spatial and temporal scales with strong and direct impact on the coastal and shelf circulation and *SST* [26,44,46]. Wind-driven coastal upwelling and Agulhas Current driven variability both create a dynamic and highly variable *SST* environment in the nearshore regions along the east coast of southern Africa. Further north, the Mozambique Channel (12–24.6° S) is characterized by the highest *SST* values within

the southern African region due to the higher stratification which occurs there [47]. The Mozambique Channel is a dynamic region where intense, southward propagating anti-cyclonic eddies are often found [48]. In addition, both cyclonic eddies and dipole eddy pairs have been observed to regularly form at the northern and southern ends of Madagascar, respectively [49,50]. Although the region is populated with numerous mesoscale features, *SST* gradients are far weaker relative to the west coast of southern Africa and inshore of the Agulhas Current [48].

The complex oceanographic environment around southern Africa and the observed discrepancies between satellite *SST* products and in-situ observations [31–33] further emphasizes the need for a regionally-focused comparison of L4 *SST* products within the southern African marine region. This study provides a detailed comparison of the freely available, operational L4 *SST* satellite data products for the southern African region by comparing the spatial and temporal discrepancies between L4 *SST* products. By highlighting the limitations of each product, this study aims to identify the most appropriate *SST* products for further regional analyses. Furthermore, this study aims to identify how both oceanographic dynamics and techniques used to produce L4 *SST* products may be responsible for the observed discrepancy between L4 products in southern Africa. The southern African region is characterized by a range of dynamically diverse oceanic provinces and the findings stemming from this study are, therefore, relevant to many other regions of the world.

## 2. Materials and Methods

The analysis included a total of 20 *SST* products with 15 Level 4 (L4) and 5 Level 3 (L3) *SST* products, as listed in Table 1, together with their respective re solutions and input data. *SST* products were only considered if the data set covered the entire southern African marine region, was operationally-available in near-real time (at the time of this study), freely available, produced daily or hourly, and was either a L3 or L4 product. L3 *SST* products refer to data sets which are spatially gridded but have missing data due to cloud cover, land contamination, sun glint or interference from aerosols such as dust [21]. The L3 product may be derived from single swaths (L3U), multiple swaths (L3C) or a combination of swaths from multiple sensors (L3S). L4 *SST* products refer to gridded and gap-free data sets. In order to obtain spatially complete fields L4 data sets merge different combinations of source data in addition to using various interpolation techniques. The majority of L4 products use various distinct techniques in order to compensate for the diurnal temperature variability between day and night satellite passes [6,51]. These techniques include excluding all daytime inputs, excluding daytime inputs under certain atmospheric conditions only, or applying various corrections to the source data [52,53]. L3 products provide information of the input data used for L4 products, and thus L3 products were used to identify regions strongly influenced by missing data and describe how this may influence the subsequently generated L4 products.

**Table 1.** Description of the various Level 4 (L4) *SST* products used in this study, including the low, medium and high-resolution *SST* products. The numbers representing the sensors used are fully expanded in Table A1 in Appendix A.

| Product | Resolution | Level | Sensors | Reference/DOI |
|---|---|---|---|---|
| AVHRR_OI*SST* | 0.25 | L4 | 1, 13 | [54] |
| GAMSSA_*SST* | 0.25 | L4 | 2, 6, 13 | [55] |
| REMSS MW_OI_ *SST* | 0.25 | L4 | 2, 3, 4, 5, 7 | [56] |
| CMEMS_GMPE | 0.25 | L4 | Ensemble of products | [57] |
| JMA MGD*SST* | 0.25 | L4 | 3, 5, 13 | [58] |
| NAVO_K10 | 0.10 | L4 | 1, 3, 8 | 1 [59] |
| CMC_*SST* | 0.10 | L4 | 1, 2, 13 | [60] |
| ODYSSEA MUR | 0.10 | L4 | 1, 2, 4, 6, 8, 9 | [61] |
| REMSS_IR_MW_OI | 0.09 | L4 | 2, 3, 4, 5, 7, 11, 10 | [62] |
| OSTIA_NRT | 0.05 | L4 | 1, 2, 3, 4, 6, 8, 9, 10, 13 | [63] |
| Geo-Polar OSPO | 0.05 | L4 | 1, 8, 10, 12, 13 | [64] |
| DMI_*SST* | 0.05 | L4 | 1, 2, 9, 10, 11 | [65] |
| ODYSSEA_AGULHAS | 0.02 | L4 | 1, 2, 3, 4, 6, 7, 9, 10 | [66] |
| G1*SST* | 0.01 | L4 | 1, 6, 8, 9, 11, 13 | [67] |
| JPL_MUR | 0.01 | L4 | 1, 2, 3, 5, 11, 13 | [68] |

Two of the 20 products, ODYSSEA_AGULHAS and SEVIRI, were regionally focused products, while the rest had global spatial coverage (Table 1). The spatial resolution of the selected *SST* products ranged from 0.25° to 0.01°. In order to account for the wide range of spatial resolutions, the products were grouped by spatial resolution into low resolution (≥0.25°), medium resolution (0.24–0.06°) and high resolution (≤0.05°) products. Each product in the respective resolution groups (low, medium and high) was re-gridded, using bilinear interpolation, onto a common spatial grid for data analysis. Low resolution (≥0.25°), medium resolution (0.24–0.06°) and high resolution (≤0.05°) products were re-gridded to spatial resolutions of 0.25°, 0.10° and 0.05°, respectively. All L3 *SST* products (Table 2) were flagged, when appropriate, to remove unreliable data such as cloud or rain contaminated data. The data analysis was conducted over a three-year period (1 January 2016 to 1 January 2019), which represented the longest common period shared by the *SST* data products across the various resolutions (Tables 1 and 2). Each of the products had a daily temporal coverage, except for the SEVIRI *SST* product which is provided at an hourly temporal resolution. The hourly SEVIRI data were thus averaged into daily means.

**Table 2.** Description of the various Level 3 (L3) *SST* products used in this study, including only high-resolution *SST* products.

| Product | Resolution | Level | Sensors | Reference/DOI |
|---|---|---|---|---|
| MODIS_TERRA (DAY/NIGHT) v2014.0 | 0.04 | L3 | MODIS on board TERRA platform | [69] |
| MODIS_AQUA (DAY/NIGHT) v2014.0 | 0.04 | L3 | MODIS on board AQUA platform | [70] |
| VIIRS (DAY/NIGHT) | 0.04 | L3 | VIIRS on board S-NPP platform | [71] |
| PATHFINDER (DAY/NIGHT) PFV53 | 0.04 | L3 | AVHRR | [72] |
| SEVIRI | 0.05 | L3 | Meteosat-11/SEVIRI | [73] |

In order to compare L4 *SST* products within the southern African region various metrics were used. The variance (V) between L4 *SST* products, as calculated per Equa-

tion (1), was used to determine both the spatial and temporal variability of the discrepancies between L4 *SST* products.

$$V = \frac{1}{N-1} \sum_{i-1}^{N} |SST_i - \mu|^2 \qquad (1)$$

where $N$ is the number of observations (*SST*) and $\mu$ is the mean of *SST*,

$$\mu = \frac{1}{N} \sum_{i=1}^{N} SST_i \qquad (2)$$

The variance across L4 *SST* products was then related to the spatial and temporal variability in the number of valid observations within various L3 *SST* products to highlight the impact of missing IR retrievals on the L4 *SST* products. A further analysis was conducted using five case studies at regions representing both high and low variance between L4 *SST* products. The Pearson correlation coefficient, standard deviation and centered root mean square error (crmse) for each high-resolution *SST* product as well as ANOVA tests were performed at each case study site to identify statistical differences between L4 *SST* products. In addition, the correlation of the variance between L4 *SST* products and *SST* gradient at each test site was calculated to determine if there was a relationship between *SST* gradient and the discrepancy across L4 *SST* products. Further explanation of the use of each metric is include in the relevant sections.

*Sub-Division of the Southern African Marine Region*

The southern African marine region is influenced by numerous oceanographic processes and features, each with distinctive characteristics. To accommodate for this diverse oceanography, the region was separated into five sub-regions (Figure 1), referred to as the northern Benguela (NB), southern Benguela (SB), Agulhas Bank (AB), East Coast (EC) and Mozambique Channel (MC) (Figure 1). These sub-regions were selected using previously published boundaries defined according to the oceanographic, atmospheric, and biological variability within each region [36,74].

## 3. Results

### 3.1. Spatial Variability between L4 Products

The variance between the observed *SST* values of each L4 *SST* product was calculated at each grid cell and for every time step over the entire region. The individual variance values were then averaged over a common three-year period (1 January 2016 to 1 January 2019), providing the mean variance per pixel across all the *SST* products. This calculation was repeated for the low, medium and high-resolution products, respectively (Figure 2-top). The variance provides a measure of the spread between the observed data at each grid point. Therefore, high variance indicates regions where the spread of observed *SST* values is large and there is high variability or disagreement among the *SST* products. Conversely, low variance indicates a small spread between observed *SST* values and hence better agreement between *SST* products. Variance maps for the low, medium and high-resolution *SST* products (Figure 2-top) all showed similar spatial patterns and magnitudes, suggesting that the variability among the *SST* products was not strongly influenced by the spatial resolution of the respective products.

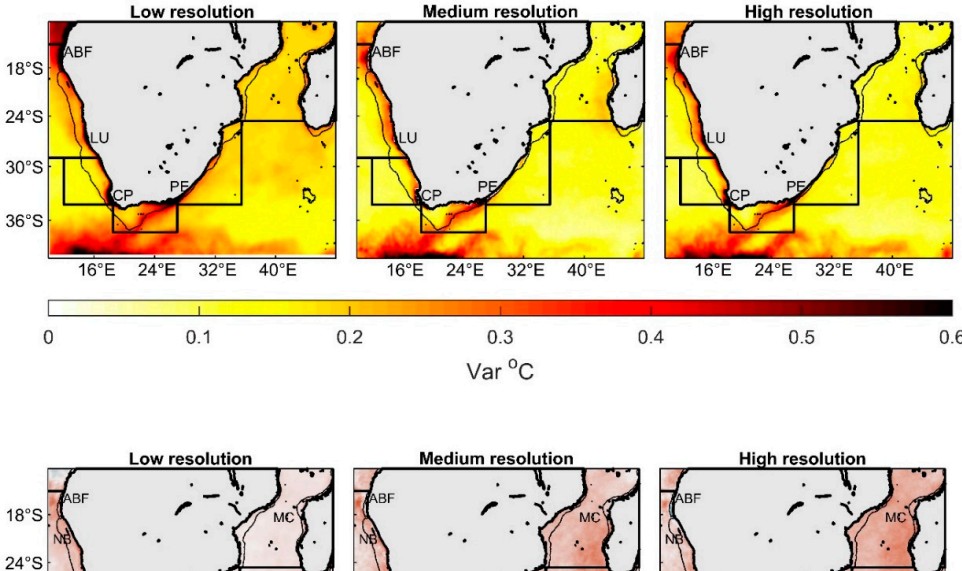

**Figure 2.** (Top panel) Variance (°C²) and (bottom panel) seasonal bias of variance (summer climatology–winter climatology), for low (0.25°), medium (0.1°) and high (0.05°) spatial resolution sea surface temperature (*SST*) products for the southern African marine region. The black boxes show the five defined sub-regions; the northern Benguela (NB), southern Benguela (SB), Agulhas Bank (AB), East Coast (EC) and Mozambique Channel (MC). The labels ABF, LU CP and PE represent the Angola–Benguela Front, Lüderitz, Cape Peninsula and Port Elizabeth regions, respectively. The thin black contour shows the 1000 m isobath.

The regions associated with highest variance, and therefore the strongest disagreement among *SST* products, corresponded to well-known oceanographic features characterized by complex temperature structures. These included the Angola Benguela Front, the Lüderitz upwelling cell, the Cape Peninsula upwelling cell, the region off Port Elizabeth, the Agulhas Current Retroflection, as well as the Agulhas Current along the Agulhas Bank. Within these regions of high variance, extremely high values (>0.8 °C²) were observed at the Lüderitz, Cape Peninsula and Port Elizabeth regions across all of the respective resolutions (Figure 2-top). Conversely, on the east coast of southern Africa, the Agulhas Current and Mozambique Channel sub-regions showed relatively low variance (<0.4 °C²), indicating good agreement between the *SST* products. Additionally, it is important to note that most of the shelf and offshore regions along the west coast of southern Africa were characterized by low variance, while the high variance areas, with the exception of the Angola Benguela frontal region, were limited to the nearshore areas coinciding with the above-mentioned upwelling cells (Figure 2-top).

### 3.2. Seasonal Bias between L4 Products

In order to assess temporal variability between *SST* products, the seasonal bias of the variance was calculated by subtracting the mean monthly climatology over the austral winter months (June, July, August) from the mean monthly climatology over the austral summer months (December, January, February). There was a strong seasonal bias (>0.5 °C²) at the Lüderitz, Cape Peninsula and Port Elizabeth regions across the various spatial

resolutions (Figure 2-bottom). The highest seasonal biases ($>0.8\ °C^2$) were observed at the Cape Peninsula upwelling cell. This conclusively shows there was increased variance, and therefore, increased disagreement between the *SST* products during the summer months in these regions. The Agulhas Current Retroflection region and the Agulhas Return Current showed a negative seasonal bias; indicating the variance between the *SST* products was highest during the winter months within these regions.

### 3.3. Influence of Missing Data within IR Retrivals

As previously mentioned, L4 *SST* products are developed through merging numerous input sources including MW retrievals, IR retrievals and at times, in situ observations. The IR retrievals are a vital source of data as they provide high spatial and temporal resolution observations. The majority of the L4 *SST* products combine data from multiple IR sensors (Table 1). *SST* fields derived from IR sensor observations contain numerous missing data points due to cloud cover, scan geometry and/or masked by quality control procedures [25]. Several L3 *SST* data sets (Table 2) were compared to investigate how missing retrievals in IR *SST* observations might drive disagreements across L4 *SST* products within the southern African region.

The percentage of valid data points of each L3 *SST* product was calculated over the common three-year period, by dividing the number of pixels with valid *SST* data by the total number of available pixels (one per time step) at each grid point over the full region. The L3 *SST* products all showed similar results and, therefore, only the MODIS TERRA results were presented in Figure 3 for brevity, while similar maps for the other L3 products have been included in the Supplementary material (Figures S1 and S2). The largest number of valid data points was observed within the Mozambique Channel, along the east coast of South Africa and in the offshore region of the west coast (Figure 3). These regions coincided with the regions of lowest variance observed among L4 products (Figure 2). Conversely, the largest number of missing data points was observed within the Angola Benguela Front, the nearshore shelf region of the northern Benguela and in the Agulhas Retroflection region (Figure 3), in agreement with the areas associated with the highest variance among the L4 products (Figure 2).

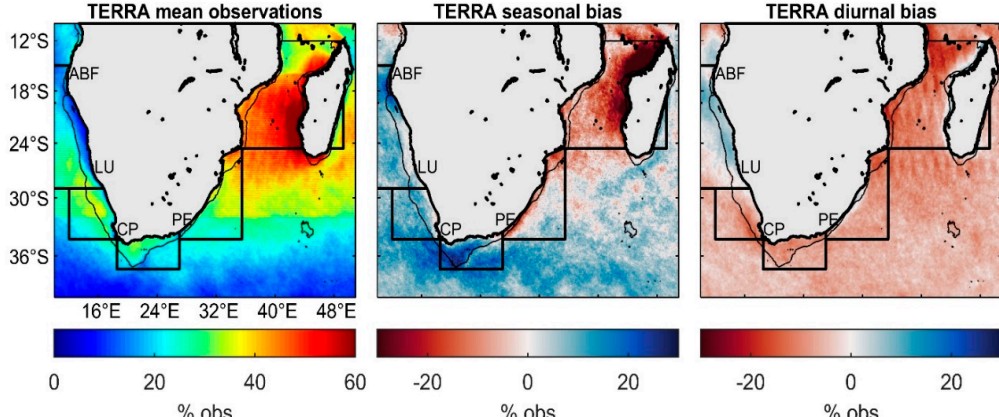

**Figure 3.** (Left panel) The mean percentage of valid observations for the L3 TERRA *SST* product during 1 January 2016–1 January 2019; (middle panel) the seasonal bias represented by the climatology of austral summer months (December, January and February) subtracted from the austral winter months (June, July and August) and (right panel) the diurnal bias represented by the day time observations subtracted from the night-time observations. The thick black boxes show the five defined sub-regions and thin black contour shows the 1000 m isobath. The labels ABF, LU, CP and PE represent the Angola–Benguela Front, Lüderitz, Cape Peninsula and Port Elizabeth regions, respectively.

The seasonal bias of the number of valid data points (climatology of valid data points for the austral winter subtracted from the valid data points for the austral summer)

showed strong seasonal differences in the offshore region of the west coast and within the Retroflection region (Figure 3). The seasonal signal in the Retroflection region again coincided with the seasonality of the variance between the L4 *SST* products (Figure 2), with larger amounts of missing L3 data corresponding to larger variance among L4 products.

The diurnal bias of valid data points was also analyzed. The bias was calculated by subtracting the percentage of valid data points of daytime observations from night-time observations (Figure 3 and Table 2). There was a negative bias across the majority of southern African region which indicated a higher percentage of valid observations in the night-time data sets. However, the northern Benguela region coastal region showed a strong positive bias and, therefore, a higher percentage of valid data point in the daytime data sets. The diurnal bias is an important observation as day and night input data is treated separately in the development of the L4 *SST* product in order to account for differences in diurnal heating.

The strong correspondence between regions of high variance among L4 *SST* products and increased missing data in IR retrievals was expected as the L4 products are reliant on either MW retrievals, in situ observations or interpolation methods during periods of missing IR retrievals. Both the MW retrievals and in situ observations have relatively lower spatial resolution than IR retrievals, and the interpolation methods used vary between the L4 products [24]. Therefore, the differences between *SST* products are most likely to be accentuated during periods of missing IR retrievals. Contrary to expectation, the regions characterized by extremely high variance values (>0.8 $°C^2$), i.e., Lüderitz, Cape Peninsula and Port Elizabeth regions, however, were not associated with substantial increases in the number of missing data points (Figure 3). This suggests the number of missing data points within IR retrievals is not the sole reason for high variance between L4 *SST* products around southern Africa.

### 3.4. Timeseries Analysis at Selected High-Variance Locations

In order to carry out further statistical and timeseries analyses, five case study sites were selected (Figure 4A). The selected regions, boxes of 13 × 13 grid cells, represent both areas of highest variance among the *SST* products (Figure 2) as well as oceanographic features of interest. These included the Lüderitz upwelling cell, Cape Peninsula upwelling cell and Port Elizabeth region. Two regions of low variance, one off the KwaZulu-Natal (KZN) Bight along the east coast of South Africa, and the other in the Mozambique Channel were included for comparison (Figure 2). The case studies are only presented for high resolution L4 products since the low and medium resolution products showed very similar patterns.

The timeseries of monthly mean *SST* (Figure 4B–D) showed there was substantially more disagreement between the L4 *SST* products at the Lüderitz, Cape Peninsula and Port Elizabeth regions compared to that in the KZN Bight and Mozambique Channel locations (Figure 4E,F). This was confirmed by Analysis of Variance (ANOVA) tests which showed that the L4 *SST* products yielded significantly different *SST* values at Lüderitz, Cape Peninsula and Port Elizabeth (Table S1). Conversely, the differences between *SST* products in the KZN Bight and Mozambique Channel regions were not statistically significant (Table S1).

The seasonal signal identified in Figure 2 was also clearly visible in the timeseries at the Lüderitz and Cape Peninsula upwelling cells with substantial disagreement between *SST* values during the austral summer months (Figure 4B,C). There were subtle variations between these case studies, with the periods of disagreement noticeably longer at Lüderitz, ranging from January to June, while the periods of disagreement at the Cape Peninsula location ranged from January to March. The Port Elizabeth region also showed increased variance in the summer months, but the seasonal differences were not as obvious, and there were more sporadic periods of increased variance throughout the year (Figure 4C). It is important to note that despite the disagreement in absolute *SST* values, the L4 *SST* products all displayed similar seasonal cycle amplitudes, as well as month-to-month variability, at the Lüderitz, Cape Peninsula and Port Elizabeth regions (Figure 4B–D). This indicates that

the large-scale variability and the seasonal cycle was fairly well represented by all the L4 *SST* products.

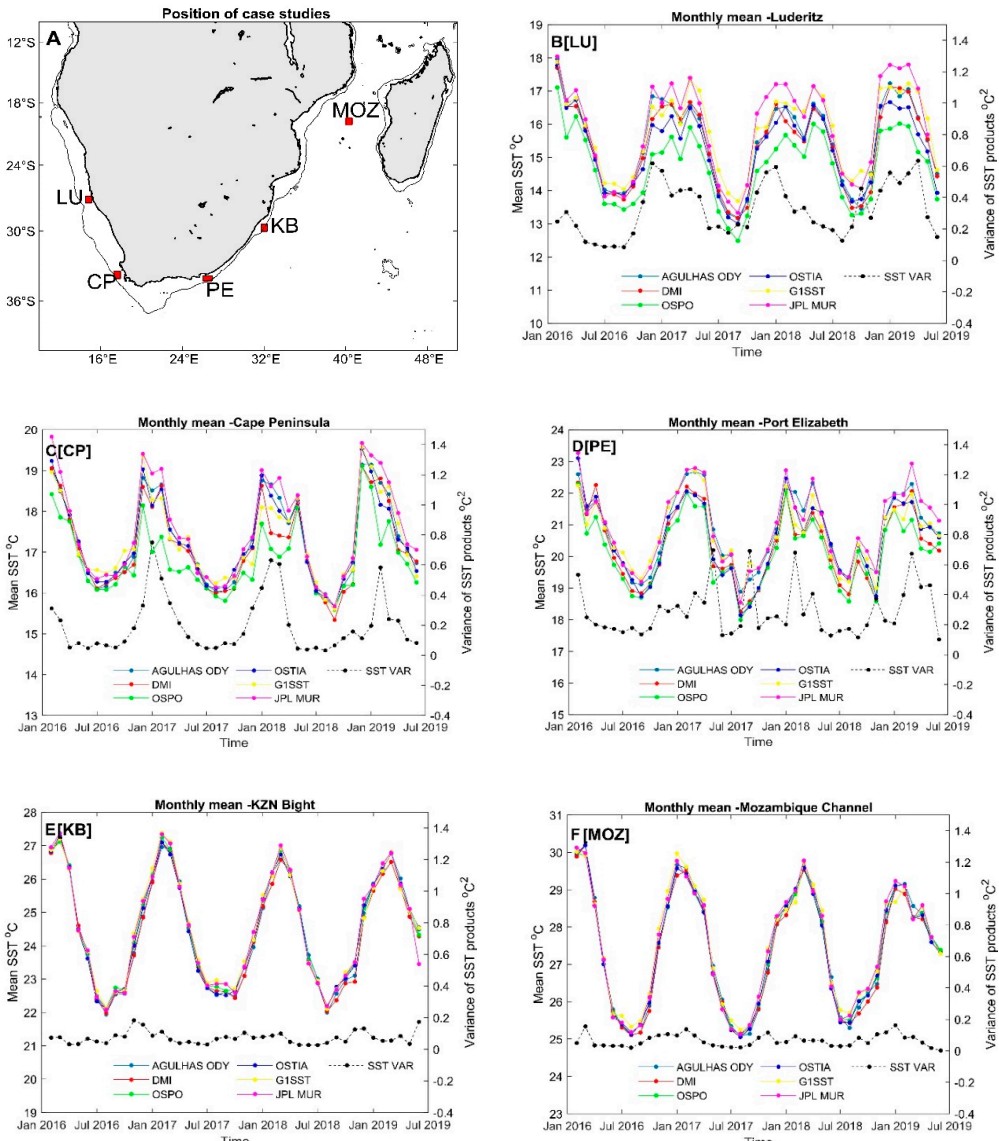

**Figure 4.** The location of the case study regions is shown in (**A**). The labels LU, CP, PE KB and MOZ in (**B**–**F**) represent the Lüderitz, Cape Peninsula, Port Elizabeth, KZN Bight and Mozambique Channel, respectively. The thin black contour shows the 1000 m isobath. Line graphs indicate monthly averaged Sea Surface Temperature (*SST*) at the selected case study regions during the 1 January 2016–1 January 2019 period. The variance of the *SST* products (black line), calculated daily and averaged monthly, is overlaid.

The JPL MUR product showed consistently higher *SST*s than the other products during periods of disagreement at the Lüderitz, Cape Peninsula and Port Elizabeth regions, with the most obvious differences occurring at the Lüderitz upwelling cell (Figure 4B–D). This is consistent with studies from the California/Baja coast where JPL MUR was shown to have a warm bias relative to other L4 *SST* products and in situ observations [75]. Conversely, the Geo-Polar OSPO *SST* product was consistently cooler relative to the other products at the Lüderitz and Cape Peninsula upwelling cells, and to a lesser extent at the Port Elizabeth region (Figure 4B-D).

The Pearson correlation, standard deviation and centered root mean square error (crmse) for each high-resolution *SST* product, calculated with respect to an ensemble

median of all the *SST* products, was presented as Taylor diagrams (Figure S3) in order to identify statistically outlying products in each of the selected regions. This statistical comparison of the *SST* products yielded similar patterns to those observed in the variance maps (Figure 2) and monthly timeseries (Figure 4B–F). Greater agreement was observed between products (i.e., small differences in the correlations, crmse and standard deviations) at selected regions on the east coast of South Africa and in the Mozambique channel, while there was less agreement between the *SST* products (larger difference in the correlations, crmse and standard deviations) at the Lüderitz, Cape Peninsula and Port Elizabeth regions (Figure S3). The Taylor diagrams indicated a more or less uniform dispersion among the products, rather than a single or group of outlying products (Figure S3). This uniform dispersion was consistent across all the low, medium and high spatial resolution products.

### 3.5. Influence of SST Gradient

While the influence of missing IR retrievals was shown to result in regional differences among L4 *SST* data sets for the southern African marine environment (Figures 2 and 3), the seasonal differences in variance observed at the Lüderitz, Cape Peninsula and Port Elizabeth regions indicated there are additional dynamics resulting in the discrepancy between L4 *SST* products within these areas. The regions of highest variance, and hence the highest variability between *SST* products, closely corresponded with regions characterized by strong horizontal *SST* gradients (Figure 5A). The mean *SST* gradient, calculated as an average from each high resolution L4 *SST* product, clearly showed the strongest *SST* gradients occurring at the Angola–Benguela Front, along the west coast of southern Africa, including both the Lüderitz and Cape Peninsula upwelling cells, in the Agulhas Current Retroflection region, as well as inshore of the Agulhas Current along the Agulhas Bank, and near the inshore edge of the Agulhas Current, close to Port Elizabeth (Figure 5A). Similar to the variance maps, the spatial distribution of *SST* gradients showed consistency across the low, medium and high-resolution products; even though there were differences in the absolute values of the *SST* gradient, with a distinct increase in *SST* gradient values with increasing spatial resolution.

Correlations were used to confirm the relationship between mean *SST* gradients and variability between *SST* products (Figure 5B–F). There were strong, significant positive correlations for the Lüderitz (r = 0.67; $p < 0.001$) and Cape Peninsula (r = 0.84; $p < 0.001$) upwelling cells. The correlation at Port Elizabeth (r = 0.35; $p < 0.05$) was much lower than those at Lüderitz and the Cape Peninsula but still statistically significant. The correlations for the KZN Bight (r = 0.16; $p = 0.32$) and Mozambique Channel (r = 0.03; $p = 0.85$) were low and not statistically significant, suggesting no relationship between *SST* variance and *SST* gradient in these areas (Figure 5E,F).

It is clear that a combination of missing IR retrievals and the strength of the horizontal *SST* gradient results in disagreement between the L4 *SST* data sets within the southern African region (Figures 2–5). In coastal upwelling regions, the number of IR retrievals is of concern as MW suffer land contamination and, therefore, are not suitable within ~50 km of the coastline [19]. Therefore, during periods of missing IR retrievals L4 *SST* products are strongly reliant on either spatially coarse in situ observations or interpolation methods.

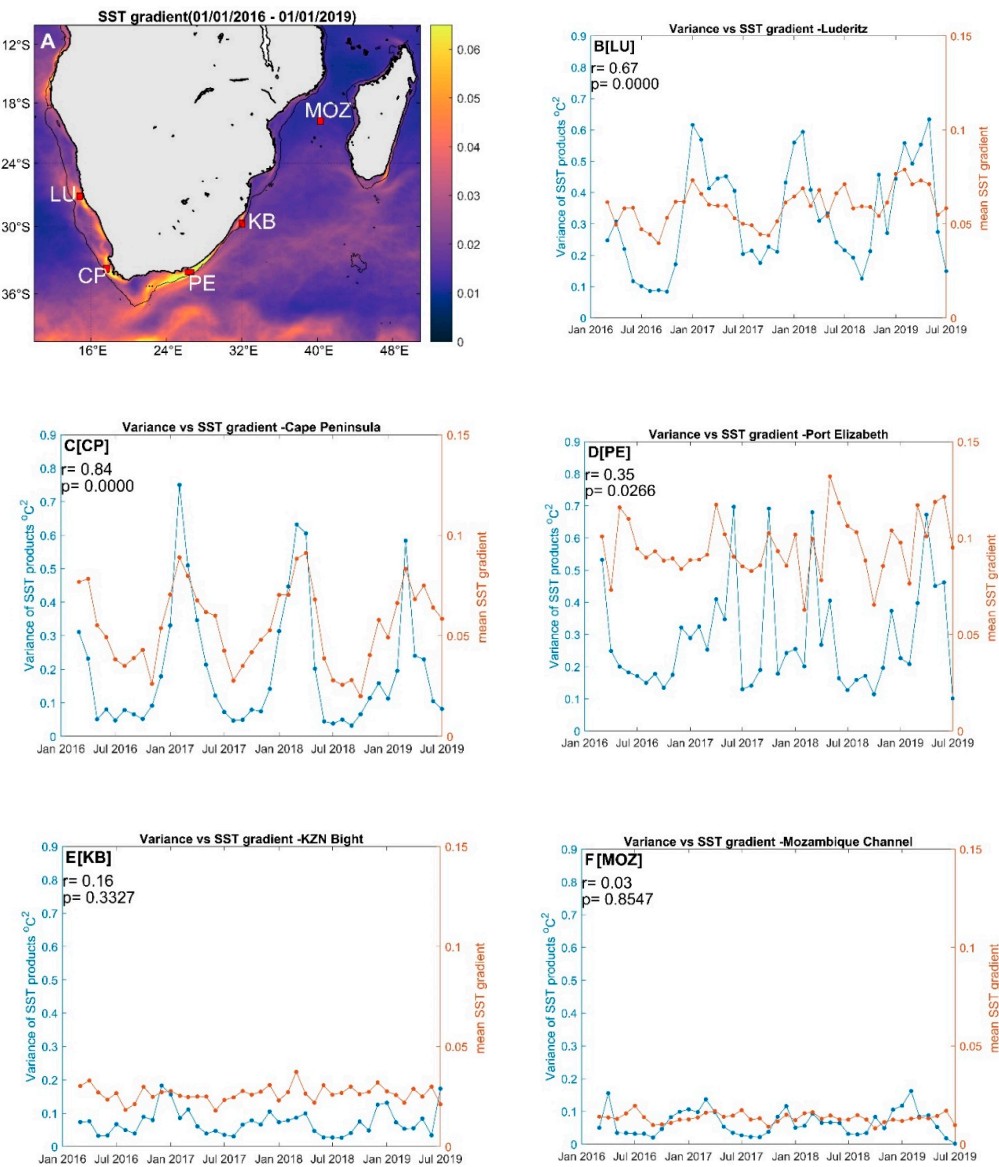

**Figure 5.** (**A**) shows the mean *SST* gradient, calculated using the high resolution L4 *SST* products, as well as the position of the case study regions. The thin black contour shows the 1000 m isobath. The remaining panels (**B–F**) show timeseries of the mean *SST* gradient (orange) and *SST* variance (blue) for the Lüderitz, Cape Peninsula, Port Elizabeth, KZN Bight and Mozambique Channel, respectively. Both the *SST* gradient and variance were calculated daily and then averaged to monthly time steps over a three-year period (1 January 2016 to 1 June 2019). The correlations (r) between *SST* variance and gradients and significance (p) are presented in the top left corner of each line graph.

Daily snapshots from the Lüderitz and Cape Peninsula coastal upwelling regions were chosen to illustrate how missing IR retrievals and strong *SST* gradients result in large variability between L4 *SST* products (Figures 6 and 7). The snapshots were chosen to represent periods of high variance between *SST* products. The medium resolution products showed similar results for each of the location (Figures S4–S7).

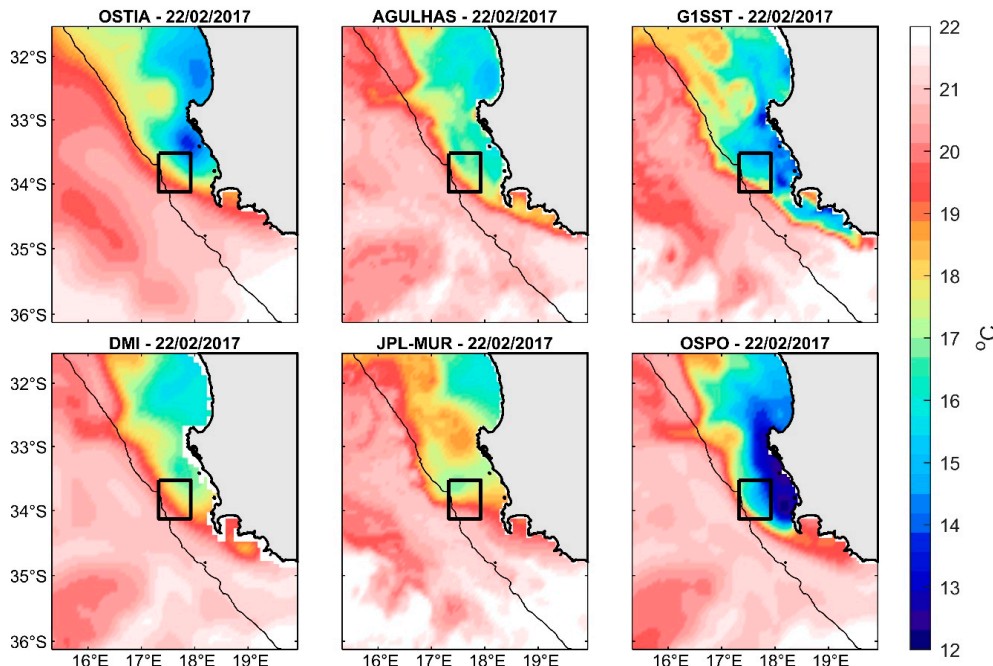

**Figure 6.** A daily snapshot for each L4, high resolution *SST* product for the 12 January 2017 in the northern Benguela. The thin black contour shows the 1000 m isobath. The black square shows the box referred to as the Lüderitz region where the highest *SST* variance was observed.

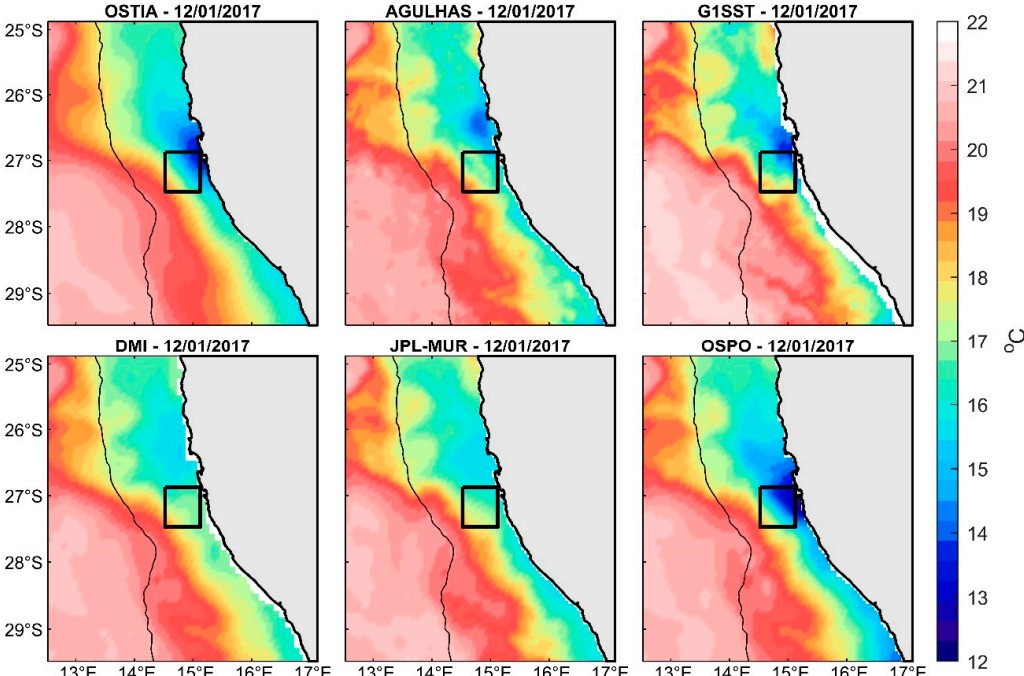

**Figure 7.** A daily snapshot for each L4, high resolution *SST* product for the 22 February 2017 in the southern Benguela. The thin black contour shows the 1000 m isobath. The black square shows the box referred to as the Cape Peninsula region, where the highest *SST* variance was observed.

Each location was associated with strong *SST* gradients, typical of upwelling events, with cool waters inshore and warmer waters offshore for the selected periods. There was reasonable agreement between the L4 *SST* products in terms of the large-scale *SST* structure (Figures 6 and 7). For example, each of the *SST* products showed a sharp offshore bend in the *SST* front at ~27° S for the Lüderitz upwelling snapshots (Figure 6). Similarly, each product observed a circular warm water feature (~32.5° S, ~17.5° E) north of the Cape Peninsula region (Figure 7).

However, there were stark differences in both the absolute values and extent of the cool inshore waters in each of the *SST* products at both locations (Figures 6 and 7). The differences in the extent of the cool inshore waters resulted in substantial differences in the position of the *SST* front among the L4 products. In both the Cape Peninsula and Lüderitz upwelling regions the Geo-Polar OSPO and OSTIA *SST* products showed substantially cooler water, up to 6 °C, inshore compared to the JPL MUR and DMI *SST* products which were characterized by much warmer inshore waters (Figures 6 and 7). The discrepancy between the JPL MUR and Geo-Polar OSPO was consistent throughout the three-year period at these locations (Figures 3 and 5).

Since MW sensors are not able to image any of the nearshore regions adequately, the discrepancies between the L4 *SST* products in these regions results from other factors, such as interpolation schemes or differences in the in-situ or IR *SST* data that they ingest. In order to identify possible causes for the discrepancy between the L4 data sets for these case studies, the L3 data sets from IR sensors were analyzed over the same period (Figures S5 and S6). For both case studies, there were substantially more IR retrievals during the daytime passes compared to night-time passes (Figures S8 and S9). Diurnal differences in IR retrievals are particularly important as selected L4 *SST* products exclude daytime data inputs to avoid the influence of diurnal heating [53]. This technique is used for both the MUR JPL and DMI *SST* products and appears to be responsible for the apparent warm bias. The daytime passes clearly showed substantially cooler inshore waters which were obscured within the night-time passes due to missing IR retrievals (Figures S6 and S9). The Geo-Polar OSPO and OSTIA *SST* products, which use both day and night-time inputs under specified wind conditions, clearly show substantially cooler inshore waters.

The influence of diurnal differences in IR retrievals demonstrated by the case studies is an important observation as there were ~20% less IR retrievals during night-time passes along the west coast of southern Africa, especially within the northern Benguela region (Figure 3). The missing IR retrieval is likely due to fog which is prevalent along the west coast of southern Africa and has been shown to occur more frequently between 00:00 and 06:00 [76,77]. Therefore, L4 *SST* products which use only night-time data in order to account for the effects of diurnal heating will have a stronger reliance on in situ observation and interpolation methods, introducing more uncertainty into the data sets, along the west coast of southern Africa.

## 4. Discussion

The increasing number and availability of L4 *SST* products emphasizes the need for comparisons between the various products, especially within the complex oceanographic environments. While large portions of the southern African marine region showed good agreement between *SST* products; there was substantial disagreement between products in regions characterized by complex and dynamic *SST* structures. These regions included the Angola Benguela Front and the retroflection of the Agulhas Current. Extremely high variance (>0.8 °C$^2$) values were observed at the Lüderitz and Cape Peninsula upwelling cells and the region off Port Elizabeth; within these regions the disagreement among the products was shown to be statistically significant. These extremely high variance values indicate that discrepancy between L4 products averaged 0.8 °C in these regions which far exceeds the accepted threshold of 0.4 K accuracy set by the GODAE High-Resolution *SST* Pilot Project (GHR*SST*-PP) for satellite *SST* products needed to improve model performance [6]. It also indicates that operational monitoring and research findings will be

strongly impacted by the choice of *SST* product in these regions. The same disagreement between *SST* products was identified across various spatial resolutions, with similar spatial and temporal variability, as well as absolute values, observed among the low, medium and high-resolution *SST* products. Therefore, simply selecting *SST* L4 products with high spatial resolution for operational or research activities is not an adequate response to address the inconsistency observed between the various L4 satellite *SST* products.

The analysis highlighted that the discrepancies between L4 *SST* products within the southern African marine region stem from the techniques used to develop the L4 *SST* products and are accentuated by the regional ocean dynamics. The discrepancies were shown to correspond to regions where the number of missing IR retrievals was highest, including the Angola–Benguela Front, the nearshore environment of the southern African west coast and the retroflection region of the Agulhas Current (Figure 3). The missing data in these regions result in each product having a greater reliance on their respective interpolation schemes, thus leading to inherent differences among the L4 products [24]. A greater reliance on interpolation schemes has been shown to over smooth *SST* fields resulting in the misrepresentation of fine scale *SST* features such as *SST* fronts [78]. This over smoothing was clearly illustrated with substantially increased variance and significant disagreement between the L4 *SST* products within the regions of strong *SST* gradient (Figures 2 and 4).

It is noteworthy that the number of missing IR retrievals may be increased in regions of strong *SST* gradients through erroneously masking retrievals. Cloud detection algorithms have been shown to misinterpret and subsequently incorrectly flag regions of steep *SST* gradient assuming cloud contamination or bad quality data [79,80]. Incorrect flagging is particularly pronounced along the west coast of southern Africa as the coastal upwelling sites are characterized by both strong *SST* gradients and large amounts of cloud cover [31–33]. Therefore, erroneous masking may lead to a greater reliance on interpolation schemes in a region of a strong *SST* gradient, and in turn leads to greater discrepancies between L4 *SST* products.

The ability of a L4 *SST* product to resolve regions of a strong *SST* gradient is further reduced within coastal regions as there are no usable MW retrievals within the coastal band (~50 km), and therefore, L4 products have a greater reliance on interpolation methods in these regions. This is evident as the largest discrepancies between the L4 *SST* products were observed at regions with strong *SST* gradients within the coastal regions; most notably at the upwelling regions of Lüderitz and Cape Peninsula, and near Port Elizabeth (Figure 4B–D). These regions provide clear examples of how the local ocean dynamics, which in turn drives the *SST* gradients, strongly influences the observed temporal variability between L4 *SST* products. Furthermore, *SST* is commonly usedto identify mesoscale structures and oceanographic fronts within the coastal band (~50 km) [44,81,82]. The increased discrepancies between L4 *SST* products within the coastal band, in regions characterized by strong *SST* gradients, is of importance to such studies and, therefore, the choice of *SST* product should be of careful consideration.

The Cape Peninsula is characterized by strong seasonal upwelling, with increased upwelling in the austral summer months resulting in stronger summer *SST* gradients [35]. The periods of strongest *SST* gradient closely matched the periods of highest variance between products at the Cape Peninsula upwelling cell with a sharp well-defined peak observed in the variance in January (Figure 4C). The Lüderitz upwelling cell was characterized by a less defined peak in variability between products extending into the austral autumn months (March, April and May), closely following the local dynamics at the Lüderitz upwelling cell, which is characterized by perennial upwelling with the strongest zonal *SST* gradients occurring in the austral autumn [40].

The observations that the greatest discrepancy between *SST* products occurs during the periods of increased upwelling within the Benguela upwelling system (i.e., off Luderitz and Cape Peninsula) correspond well with studies in the Peru and Iberian upwelling systems where the performance of *SST* products was shown to decrease with increasing

upwelling intensity [30]. This link is an important consideration as upwelling events drive large amounts of productivity and therefore, are of interest for both operational and research activities [37]. The disagreement between *SST* products will strongly impact monitoring and research focusing on upwelling dynamics, as well as model validation and assimilation within these regions. Therefore, careful consideration is needed when using satellite *SST*s in these regions especially when focusing on the event scale (relatively short temporal and spatial scales). To compensate for the observed discrepancies in the region, the use of ensemble means and testing the sensitivity of outputs using several *SST* products is strongly suggested; ideally with in situ observations being used to supplement the satellite observations.

The variance observed at Port Elizabeth was not associated with a clear seasonality. The Agulhas Current inshore front is generally found near the 1000 m isobath offshore of Port Elizabeth [26]. This region of a strong *SST* gradient is subject to a range of meso and sub-mesoscale processes such as Agulhas Current meanders, shear-edge eddies, filaments or warm water plumes which occur throughout the year. Closer to the coast off Port Elizabeth, wind-driven coastal upwelling is frequently observed during summer [83,84], and rapid southward advection of this cooler upwelled water along the inshore edge of the Agulhas Current may further contribute to the strong *SST* gradients and large variability observed. Resolving the high spatial and temporal variability which characterizes the Agulhas Current frontal region using satellite-based observations remains a significant challenge and explains the lack of agreement between *SST* products in the region (Figure 4D). The much higher temporal and spatial variability of this region is to likely be responsible for a relatively lower correlation between the monthly averaged *SST* gradient and variance between L4 *SST* products (r = 0.35), compared to the Lüderitz (r = 0.67) and Cape Peninsula (r = 0.84) upwelling locations (Figure 5B–D).

The comparisons showed that periods of increased disagreements between L4 *SST* products were characterized by a larger spread among *SST* values, rather than a single, or group of outlying products (Figure 4B–F and Figure S3). Similar findings were reported by when comparing L4 *SST* products to in situ temperature loggers in the Chilean and Iberian upwelling region [30]. The decrease in the performance with increasing upwelling intensity was shown to be pervasive among L4 *SST* products, rather than driven by individual poorly performing products [30]. Validation of L4 *SST* products within the California/Baja upwelling system also showed similar results with little statistical differences between the respective L4 products [75], and each of the products in the region were strongly correlated with in-situ observations.

Finally, the techniques used to account for diurnal heating may increase the discrepancy between L4 *SST* products along the west coast of southern Africa. Several L4 *SST* products exclude daytime data inputs to account for diurnal heating. However, the west coast of southern Africa is characterized by increased fog between 00:00 and 06:00 [76,77], which will result in decreased IR retrievals (Figure 3) and, therefore, a greater reliance on the respective interpolation schemes. This effect is highlighted in the case studies (Figures 5 and 6); with L4 *SST* products using only night-time inputs observing smaller upwelling events and showing a warm bias relative to the other L4 *SST* products at the Lüderitz and Cape Peninsula upwelling cells (Figure 4B,C). The bias towards warmer inshore waters is likely due to the increased weight of offshore pixels during the interpolation process which is expected to dampen the coastal upwelling signal [30]. Further validations using in situ observations are needed to quantify the bias introduced from using only night-time data within this region.

While this study clearly highlights the discrepancies between L4 *SST* products in regions of strong, highly-variable *SST* gradients, future studies using in situ comparisons with L4 *SST* products are needed in order to distinguish and quantify which interpolation scheme and development techniques are most suitable for the southern Africa coastal upwelling regions. The techniques used to compare L3 products to in situ observations along the southern Africa coastline may be extended to include L4 *SST* products [33].

Similarly, future strategies used to compare L4 *SST* products with in situ data in the Southern African region, may follow from what has been done within the Chilean, Iberian and Baja upwelling systems [30,75].

## 5. Conclusions

While there was good agreement between *SST* products for most of the southern African marine region, there was substantial disagreement in regions characterized by a strong *SST* gradient, particularly in highly dynamic regions and within coastal upwelling systems. This disagreement will strongly impact research and operational outputs in these regions, although studies focusing over large spatial and temporal scales will be less affected as the seasonality and large-scale *SST* structures showed good agreement among products. Our findings are in agreement with previous studies [30,75] and highlight the pervasive nature of the disagreement between products, independent of spatial resolution, in dynamic regions of intense upwelling and strong *SST* gradients. The notion of identifying a "superior" *SST* product to improve research and monitoring outputs is, therefore, misplaced in these regions. Instead, this study highlights the need to better exploit the daytime *SST* acquisitions in coastal upwelling regions and the need for dedicated networks of in situ observations, with high frequency temporal resolution sampling to supplement the satellite products. This would improve our ability to effectively observe and monitor as well as develop robust modelling products for the ecologically and economically important regions within the Southern African marine region.

**Supplementary Materials:** The following are available online at https://www.mdpi.com/2072-429 2/13/7/1244/s1, Figure S1: The mean percentage of missing observations of the L3 *SST* products from 1 January 2016–1 January 2019, Figure S2: The mean seasonal bias of missing observations as a percentage of the L3 *SST* products from 1 January 2016–1 January 2019, Figure S3: Taylor Diagrams displaying the Pearson correlation coefficient, standard deviation and centred root mean square error (crmse) for the high-resolution Level 4 (L4) *SST* products., Figure S4: A daily snapshot for each Level 4 (L4), medium resolution *SST* product for the 12 January 2017 in the northern Benguela., Figure S5: A daily snapshot for each Level 3 (L3) *SST* products for the 12 January 2017 in the northern Benguela, Figure S6: A daily snapshot for each night-time L3 *SST* product for the 12 January 2017 in the northern Benguela, Figure S7: A daily snapshot for each Level 4 (L4), medium resolution *SST* product for the 22 February 2017 in the southern Benguela, Figure S8: A daily snapshot for each Level 3 (L3) *SST* product for the 22 February 2017 in the southern Benguela Figure S9: A daily snapshot for each night-time Level 3 (L3) *SST* product for the 22 February 2017 in the southern Benguela, Table S1: Results of the ANOVA tests conducted to compare the high resolution L4 *SST* products at the Lüderitz, Cape Peninsula, and Port Elizabeth regions.

**Author Contributions:** Conceptualization, T.L.; methodology, T.L. and M.C.; software, T.L.; validation, M.C.; formal analysis, M.C.; investigation, M.C. and T.L.; resources, T.L.; data curation, M.C.; writing—original draft preparation, M.C.; writing—review and editing, T.L and M.K.; visualization, M.C.; supervision, T.L. and M.K.; project administration, T.L.; funding acquisition, T.L. and M.K. All authors have read and agreed to the published version of the manuscript.

**Funding:** This research was funded by South African Department of Environment, Forestry, and Fisheries (DEFF) and South African Environmental Observation Network (SAEON).

**Acknowledgments:** The Oceans and Coasts Research component of the South African Department of Environment, Forestry, and Fisheries (DEFF), the South African National Oceans and Coasts Information Management System (OCIMS), as well as the South African Environmental Observation Network (SAEON) are thanked for administrative support and facilities, and for funding this work.

**Conflicts of Interest:** The authors declare no conflict of interest.

## Appendix A

**Table A1.** Description of the sensors used to construct the L4 *SST* products.

| Reference Number | Sensor |
| --- | --- |
| [1] | AVHRR |
| [2] | AMSR2 |
| [3] | AMSR-E |
| [4] | TMI |
| [5] | WindSat |
| [6] | AATSR |
| [7] | GMI |
| [8] | GOES |
| [9] | SEVIRI |
| [10] | VIIRS |
| [11] | MODIS |
| [12] | JAMI |
| [13] | In situ |

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
