# Peer review of "Satellite Sea Surface Temperature Product Comparison for the Southern African Marine Region"

_remotesensing, doi:10.3390/rs13071244_

Round 1

Reviewer 1 Report

REVIEW REPORT FOR

“Satellite Sea Surface Temperature product comparison for the southern African marine Region”

By Corr et al.

In this paper the authors investigate the performances of several L4 SST products in the south African region, with the aim of testing their potential in future monitoring/operational applications. To me, the paper is clear and very well written, the figures are well presented and the English is clear. I acknowledge the authors for their work. I only have some comments/suggestions which in my opinion could improve the manuscript before the publication.

General Comments

  • I think this paper is lacking some comments which you could probably insert in your introduction or discussion section: what type of monitoring/operational applications are identified as high priority tasks for South Africa? Which is the main application you aim to develop relying on L4 SST data?
  • In this work ,you perform inter-comparisons of satellite L4 SST data. In the final section you mention future comparisons with in-situ data. Could you please add some lines indicating how you plan to perform these future analyses?
  • Can the authors comment on the possibility of relying on L3 SST data for their future monitoring applications? Do you necessarily need L4 processing level?

Specific comments

Abstract

  • Line 16: I suggest to change “15 merged” with “Several satellite-derived”
  • Line 16: I suggest to change “those best suited” with “their potential”
  • Line 24: After reading the manuscript I could understand what you mean by “Disagreement among products was consistent across spatial resolutions, indicating that spatial resolution did not strongly influence performance of the merged SST products”. However, I think this should be rephrased as “disagreement among products showed little dependence on their spatial resolutions” or something similar. Personally I struggled a bit to get the meaning of the sentence in the context of the abstract.
  • Line 30: what do you mean by high SST gradient? Larger than? Could you put a threshold?

Introduction

  • Line 46: I suggest to put two additional references on the importance of using SST for research/applications studies. Ref1, Ref2(see References Section at the end of this report)
  • Line 57: Please add Ref 3
  • Line 59: Radio Frequency Interferences are another source of uncertainty for MW retrievals
  • Line 64: Please try to provide more details on the meaning of L4 / L3 data. Not all the readers are necessarily familiar with this nomenclature. Moreover, a L4 SST product cannot be simply defined as “merged”. A L3 product can also be obtained merging SST observations from several observations/sensors (e.g. L3Collated, L3Supercollated, L3C/L3S). I suggest to rephrase the line.
  • Line 69: mostly a question:” which L4 SST data include model outputs in the L4 analyses? Or maybe you simply mean that some L4 SST products are based on numerical simulations?” I think it is crucial to make a distinction or to clarify
  • Line 74: Please add Ref3 again

Materials and Methods

  • Overall comment: the methods description seems to be lacking, you only give information on the metrics of your comparative study at section 3. I think you should add some lines on the methods and metrics of your comparison in this section (even providing explicit expressions of the equations you have been using to carry out the SST inter-comparisons, e.g. variance,…)

  • Line 131: I am not sure I understand the expression “was still active”. Do you mean that the operational service was still ongoing? IF so, it sounds a bit awkward, you should probably rephrase this
  • Line 139-140: see again my comments of line 64. L3 data can also include data from multiple sensors and still have gaps (L3S)
  • Line 151: can you please provide few words on the regridding method?

Results

  • Lines 272,275: °C2 should be changed to °C2 (also double check throughout the manuscript)

Discussion

  • Lines 526,529: I am not sure I have understood these lines. It seems you are comparing a °C2 unit with a °C unit. Please try to be more specific

    References

Ref 1 Rio, M. H., Santoleri, R., Bourdalle-Badie, R., Griffa, A., Piterbarg, L., & Taburet, G. (2016). Improving the altimeter-derived surface currents using high-resolution sea surface temperature data: a feasability study based on model outputs. Journal of Atmospheric and Oceanic Technology, 33(12), 2769-2784.

Ref 2 Ciani, D., Rio, M. H., Menna, M., & Santoleri, R. (2019). A synergetic approach for the space-based sea surface currents retrieval in the Mediterranean Sea. Remote Sensing, 11(11), 1285.

Ref 3 Robinson, I. S. (2004). Measuring the oceans from space: the principles and methods of satellite oceanography. Springer Science & Business Media.

Author Response

Dear Reviewer,

We wish to thank you for the time taken to review this paper. We have responded to each comment and have edited the original manuscript accordingly. The original text of the reviewer’s responses is in black text and the authors responses is in red text. Each time an edit to the manuscript was required a quote of the edit and corresponding line number is included in the response document.   

See the attached below 

Regards,

Matthew Carr (Corresponding author)

Reviewer 2 Report

The manuscript by Carr et al. submitted to Remote Sensing, entitled “Satellite Sea Surface Temperature product comparison for the southern African marine region” evaluates 15 sea surface temperature (SST) products for the southern African marine region. They found a general good agreement among the products, but differences were found in areas with high SST gradient or local oceanographic features, and also bias related to sampling and regional dynamics. To overcome these limitations, they suggest the use of ensemble means. The study is relevant to oceanographic and climatic studies in the southern African marine region as it evaluates SST products. The manuscript is well written, but I think it requires some improvements which I detail below.

Minor comments

Please capitalize the word “current” when it refers to a specific current (e.g., line 107, 109).

Line 107 – The Agulhas Current system is referred as it is in Figure 1, but there is no indication of the currents in Figure 1.

Line 129 – “The analysis included 20 SST products”- in the abstract, you say 15 products were analyzed. Please clarify this issue in the text, by mentioning the use of L3 and L4 products.

Tables 1 and 2 – Instead of just citing the webpage were the data is available, it would be interesting to include a column with the name of the institution responsible for making the data available (or maybe even remove the column containing the website, keeping just the institution name).

Lines 223/228/273/279 – Please refer that the variance results are in “Figure 2 – top”.

Figure 2 – Bar title: Please rewrite “Var (°C)”.

Lines 272/275/284/286 and others: Please rewrite the squared degree correctly: “°C2

Line 285 – Please refer that the variance results are in “Figure 2 – bottom”.

Figure 3 – The title of the first panel contains an error: rewrite “observations”.

Section 3.4. Title: “Timeseries”

Figures 4 and 5 – It would be easy to refer to Figures 4 and 5 subfigures by adding letters (e.g., a), b), c), etc.) to each one of them, instead of just citing their position. It is also hard to make a correspondence between the case study points showed in the first box of the Figure and the other boxes containing the data because there is no indication of the acronyms in the Figure. Make the correspondent changes also in the main text.

Figure 5, first box: please put the letters indicating the case studies in a contrasting color, e.g., white, because they are not readable.

Line 405 – I would like to see the Taylor diagrams in the main text instead of in the Supplementary Material. It would complement the comparison of the products.

Author Response

Dear Reviewer,

We wish to thank you for the time taken to review this paper. We have responded to each comment and have edited the original manuscript accordingly. The original text of the reviewer’s responses is in black text and the authors responses is in red text. Each time an edit to the manuscript was required a quote of the edit and corresponding line number is included in the response document.    

Please see the attachment below.

Regards,

Matthew Carr (Corresponding author)

Reviewer 3 Report

Comments to authors

I congratulate the authors for the work done and for their contribution to oceanographic analysis using this type of variable. I only made some observations that I think should be specified in the writing for a better understanding of the oceanography of the study area and to add an analysis that is important in the methodology and results.

Introduction:

Lines 43-50. Specify what type of oceanographic processes can be identified and studied by means of these SST images and mention some examples of these studies. Likewise, antecedents of studies mentioning the relationship between environment, resource and effects on the ecosystem as mentioned in line 49, to reinforce the application of the analysis carried out by the authors.

Lines 52-66: Apply the antecedents a little to describe the aforementioned studies since it is important to highlight them in the text to improve the content of this paragraph and not only go to consult the referecnias as readers we are interested in the author developing these antecedents.

Lines 76-83: as well as the studies on upwelling are mentioned, those related to ocean fronts caused by said upwelling should be added, effects on the ecosystem, studies of trends and identification of warm and cold periods that affect the ecosystem as a reference and use and justification for the analysis proposed by the authors.

Lines 85-99: Very good explanation ... in this sense, each paragraph of the introduction should go.

Lines 106-115: develop what type of eddies appear and their effects since each one has different characteristics and effects in the water column. Merely mentioning mesoscale structures is not enough, the type of sturcture must be mentioned.

Lines 117-127: In addition to the analysis carried out by the authors, try to mention some studies with high resolution images of SST, such as those carried out in the California Current and the Gulf of California, and to be able to give a recommendation for studies of these mesoscale structures. in the area near the coast (within 50 km). These studies are easy to obtain in articles published in the special volumes of TSM in the journal Remote Sensing.

Materials and Methods:

Lines 162-167: I recommend adding some short analysis on the identification of mesoscale structures mentioned and adding a small table with the frequency and duration of these oceanographic structures.

Results:

Good job with the results shown.

Please improve the resolution of figures 3, 4 and 5. or, where appropriate, separate and make the graphs of the larger time series for a better visualization.

Discussion:

In the sense that the suggestions of the reviewers are added in the introductory part and methodology, modify or add what is convenient in the discussion.

Author Response

(The authors gave the same response as above.)
